**Data Availability Statement:** All relevant data are within the manuscript and its Supporting Information files.

# Comparative effectiveness and cost-effectiveness of three first-line EGFR-tyrosine kinase inhibitors: Analysis of real-world data in a tertiary hospital in Taiwan

**Szu-Chun Yang** [1,2], **Wu-Wei Lai** [3], **Jason C. Hsu** [4], **Wu-Chou Su** [1], **Jung-Der Wang** [1,2]*

**1** Department of Internal Medicine, National Cheng Kung University Hospital, College of Medicine, National Cheng Kung University, Tainan, Taiwan, **2** Department of Public Health, College of Medicine, National Cheng Kung University, Tainan, Taiwan, **3** Department of Surgery, National Cheng Kung University Hospital, College of Medicine, National Cheng Kung University, Tainan, Taiwan, **4** School of Pharmacy and Institute of Clinical Pharmacy and Pharmaceutical Science, College of Medicine, National Cheng Kung University, Tainan, Taiwan

* jdwang121@gmail.com

## Abstract

### Introduction

Comparison of the effectiveness and cost-effectiveness of three first-line EGFR-tyrosine kinase inhibitors (TKIs) would improve patients' clinical benefits and save costs. Using real-world data, this study attempted to directly compare the effectiveness and cost-effectiveness of first-line afatinib, erlotinib, and gefitinib.

### Methods

During May 2011-December 2017, all patients with non-small cell lung cancer (NSCLC) visiting a tertiary center were invited to fill out the EuroQol five-dimension (EQ-5D) questionnaires and World Health Organization Quality of Life, brief version (WHOQOL-BREF), and received follow-ups for survival and direct medical costs. A total of 379 patients with EGFR mutation-positive advanced NSCLC under first-line TKIs were enrolled for analysis. After propensity score matching for the patients receiving afatinib ($n = 48$), erlotinib ($n = 48$), and gefitinib ($n = 96$), we conducted the study from the payers' perspective with a lifelong time horizon.

### Results

Patients receiving afatinib had the worst lifetime psychometric scores, whereas the differences in quality-adjusted life expectancy (QALE) were modest. Considering 3 treatments together, afatinib was dominated by erlotinib. Erlotinib had an incremental cost-effectiveness of US$17,960/life year and US$12,782/QALY compared with gefitinib. Acceptability curves showed that erlotinib had 58.6% and 78.9% probabilities of being cost-effective given a threshold of 1 Taiwanese per capita GDP per life year and QALY, respectively.

**Funding:** We are grateful for the following grants from the Ministry of Science and Technology (MOST107-2320-B-006-069 and MOST108-2918-I-006-006 to SCY) and National Cheng Kung University Hospital NCKUH-10902047 to SCY). The study sponsor had no interference with the study design, data collection and analysis, decision to publish, or preparation of the manuscript.

**Competing interests:** The authors have declared that no competing interests exist.

**Abbreviations:** EGFR, epidermal growth factor receptor; EQ-5D, EuroQol five-dimension; ICER, incremental cost-effectiveness ratio; NHI, National Health Insurance; NSCLC, non-small cell lung cancer; QALE, quality-adjusted life expectancy; QALY, quality-adjusted life year; QoL, quality of life; RECIST, Response Evaluation Criteria in Solid Tumors; T790M, substitute mutation of threonine (T) with methionine (M) at position 790 of exon 20; TKI, tyrosine kinase inhibitor; WHOQOL-BREF, World Health Organization Quality-of-Life—Brief Version.

## Conclusion

Erlotinib appeared to be cost-effective. Lifetime psychometric scores may provide additional information for effectiveness evaluation.

## Introduction

In Asian countries such as Japan and Taiwan, more than half of non-small cell lung cancer (NSCLC) patients tested for epidermal growth factor receptor (EGFR) mutations have shown positive results [1]. In addition to new generation osimertinib [2], three EGFR-tyrosine kinase inhibitors (TKIs)–afatinib, erlotinib, and gefitinib—are commonly used as first-line therapies for advanced NSCLC. Although a randomized trial showed afatinib is superior to first-generation TKIs in progression-free survival [3], a significant difference in overall survival has not been revealed [4]. Based on our clinical observation, the quality of life (QoL) and costs among patients receiving different EGFR-TKIs may differ. To improve patients' clinical benefits and save costs, the comparative effectiveness and cost-effectiveness of these drugs warrant further exploration.

Previous studies comparing the effectiveness and cost-effectiveness of first-line erlotinib versus gefitinib, erlotinib versus afatinib, and afatinib versus gefitinib, usually used model analyses and trial data [5–7]. Constructing the model analyses requires several assumptions, and utility values of QoL are often borrowed from other investigations. Although trial data are generally cleaner, their restrictive inclusion and exclusion criteria and limited length of follow-up period may limit the application in daily practice. A cost-effectiveness study using real-world approach would be useful in assisting healthcare resources allocation. Moreover, most previous analyses used chemotherapy as the reference group for indirect treatment comparisons [5, 6].

From May 2011 to December 2017, we prospectively invited all lung cancer patients visiting a tertiary center to provide their survival, QoL, and costs data for analysis. By integrating the long-term survival with utility values of QoL and costs, we developed a method to estimate the quality-adjusted life expectancy (QALE) and lifetime costs. Because psychometric scores are more sensitive than the utility values [8] and may provide additional information for effectiveness evaluation, lifetime psychometric scores were also estimated. Using the new method and real-world data of a tertiary hospital in Taiwan, this study attempted to directly compare the effectiveness and cost-effectiveness of three first-line EGFR-TKIs.

## Methods

This study was approved by the Institutional Review Board of National Cheng Kung University Hospital (NCKUH) before commencement (A-ER-107-107). All participants provided written informed consent. We performed the study from payers' perspective, and the time horizon was lifelong.

From May 2011 to December 2017, we invited all lung cancer patients who visited the outpatient departments of NCKUH to fill out QoL questionnaires, and receive follow-ups for survival and healthcare expenditures. Throughout 2017, we also recruited patients from the thoracic ward. There were 729 patients with EGFR mutation-positive advanced NSCLC under first-line TKIs during the study period. After excluding patients without informed consent and cases with missing values on EuroQol five-dimension (EQ-5D) questionnaires, all subjects were included for analysis. More specifically, the QIAamp DNA Mini Kit (Qiagen, Valencia, CA, USA) was used to analyze EGFR mutations of effusion cytology and tissue samples. We excluded patients with tumor stages I, II, and IIIA at the initiation of EGFR-TKIs, leaving only subjects with

recurrent or newly-diagnosed advanced NSCLC in the analysis. Afatinib, erlotinib, and gefitinib [9] were defined as the standard first-line therapies because osimertinib [2] had not yet become a standard therapy during the study period.

## Propensity score matching

We created a system to abstract age, sex, performance and recurrence statuses at the initiation of therapy from electrical medical records. Because all these data are required to be approved for receiving the first-line EGFR-TKIs in our hospital, the information collected were relatively complete, leaving few patients with missing performance statuses. In addition, we reviewed the reports of brain magnetic resonance imaging and computed tomography with contrast to define brain metastasis. That is, subjects who did not receive brain images or show any radiographic evidence at the initiation of therapy were categorized as negative for brain metastasis.

To account for observed covariates among three different EGFR-TKIs, we used propensity score matching via greedy algorithm [10]. That is, the first treated unit was selected to find its closest control based on the difference of their propensity scores using logistic regression. The procedure was repeated for all the treated units. We first matched patients receiving afatinib versus erlotinib one-to-one, followed by one-to-two matching for gefitinib versus erlotinib. Previous literatures have found performance status, recurrence status, metastasis, and mutation subtypes to be prognostic factors of survival [11] and QoL outcomes [8, 12] among patients receiving first-line EGFR-TKIs. Therefore, we computed propensity scores by age, sex, and these clinical characteristics upon initiation of treatment. The balances between afatinib, erlotinib, and gefitinib were tested using standardized differences, an absolute value less than 0.1 suggests each two groups are well balanced.

## Effectiveness

Each matched patient underwent follow-ups from the initiation of EGFR-TKI until September 2018 to verify the survival status. By using a semiparametric method explained in detail in our previous article [13], we extrapolated the survival to lifetime to estimate the life expectancy of patients receiving one of three first-line treatments. The extrapolation method has been shown to be effective via computer simulations [14], mathematical proof [15] and corroboration by examples of lung cancer cohorts [13, 16, 17]. The iSQoL statistical package (www.stat.sinica.edu.tw/isqol/) was used to perform the computations.

A thoracic oncologist independently reviewed every chest computed tomography, bone scan, positron emission tomography, and brain image to determine if there is any disease progression. Disease progression was defined according to the Response Evaluation Criteria in Solid Tumors (RECIST) version 1.1 [18]. Switching of therapy due to adverse events without an image progression was not considered as disease progression.

## Quality of life

The EQ-5D and World Health Organization Quality-of-Life—Brief (WHOQOL-BREF) questionnaires were used to estimate the QoL utility values and psychometric scores, respectively. We invited patients to fill out the questionnaires each time they visited our hospital to capture dynamic changes in their QoL along the follow-up course (i.e., QoL at different stages of the disease). To minimize collinearity, repeated measurements were performed more than 2 weeks apart. Using the EQ-5D scoring function from Taiwan [19], we transformed the health state parameters into a utility value ranging from 0 to 1, where 0 represents death and 1 indicates full health. To present the utility value for each group, we constructed linear mixed models to consider random effects from subjects because EQ-5Ds were repeatedly measured. The

intercept represents the mean utility value. Each facet in the WHOQOL-BREF was scored from 1 to 5, where a higher score indicated a better QoL [20]. By multiplying the average of the scores of all facets in the same domain by four, a domain score was also calculated, ranging from 4 to 20. High correlation coefficients between Rasch scores and the crude domain scores have been documented [21], which supports the WHOQOL-BREF as a sound instrument to measure QoL for cancer patients.

The time after treatment for each QoL measurement was defined as the period between the initiation of treatment and that of the interview. We used kernel-smoothing (i.e., a moving average of the nearby 10%) to estimate the mean QoL function after initiation of treatment [22]. The QoL scores beyond the follow-up period were assumed to be the same as the average of the last 10% near the end of follow-up. We multiplied lifetime survival function by the mean QoL functions of EQ-5D and WHOQOL to obtain quality-adjusted survival curves, with the sum of the areas under the curves being the QALE and lifetime psychometric scores, respectively. We applied a 3% annual discount when the QALE was employed in the estimation of the incremental cost-effectiveness ratio (ICER).

## Medical costs

We used the reimbursement database at NCKUH to obtain spending details from the initiation of treatment to December 2017. These data included all expenditures reimbursed by National Health Insurance (NHI) plus out-of-pocket money paid to the hospital, of which direct medical costs along time course of the disease could be obtained. Specifically, the total monthly healthcare expenditures were divided by the effective sample sizes, i.e., the number of patients who survived in that month, to obtain the average monthly healthcare expenditures per case. Similar to the estimation of QALE, costs beyond the follow-up period were assumed to be the same as the average of the last 10% near the end of follow-up. These values were subsequently multiplied by the corresponding monthly survival probabilities and summed to obtain the lifetime costs per case. All payments in different calendar years were adjusted based on the related consumer price indices and made them equivalent to 2017 dollars. To discount costs in future years, an annual discount rate of 3% was applied.

We did not collect transportation costs, payments to caregivers, or home adaptations due to illness or human capital loss in this analysis. The costs of EGFR-TKIs were stratified according to the order codes established by the NHI. The costs of chemotherapies included different-line regimens and administration fees.

## Probabilistic sensitivity analysis

We assumed a normal distribution for life expectancy and QALE [23], and a gamma distribution for costs, with means and standard deviations set to base-case values. To determine the most cost-effective option using net life years or QALY gained, a Monte Carlo simulation with 1,000 iterations was conducted to construct acceptability curves of the three different EGFR-TKIs. We adopted the criterion suggested by WHO-CHOICE (World Health Organization-CHOosing Interventions that are Cost Effective), and applied one gross domestic product (GDP) per capita as the threshold for cost effectiveness. Cost-effectiveness scatter plots were also developed. SAS 9.4 and Amua 0.2.7 were used to perform the analyses.

## Results

From May 2011 to December 2017, a total of 729 patients in NCKUH received afatinib, erlotinib, or gefitinib as first-line therapies for EGFR mutation-positive advanced NSCLC. Among them, 346 cases did not sign the informed consent and 4 cases with missing values on EQ-5D

questionnaires, leaving 379 subjects (**Fig A** in S1 Appendix). After 1:1:2 (afatinib: erlotinib: gefitinib) propensity-score matching the patients, 192 patients were used for analysis. Table 1 shows the 192 propensity score-matched patients stratified by treatment as well as those excluded after matching. The daily prices per person were US$48.2, US$37.0, and US$36.3 for 40mg afatinib, 150mg erlotinib, and 250mg gefitinib, respectively. In general, propensity-score matched patients had higher proportions of men, brain metastasis, and common mutations compared with those without matching. After propensity score matching, most of the characteristics in the three groups were balanced. The progression-free survival and overall survival under three first-line treatments were also similar (**Fig B** in S1 Appendix). Patients receiving erlotinib had a higher mean utility value compared with the afatinib and gefitinib groups.

## Base case scenarios

Fig 1 depicts the cost- and quality-adjusted survival curves along time courses for different treatments. In this figure, the survival probability was multiplied by the costs and QoL at each time point $t$ (see **Fig C** in S1 Appendix for the mean QoL curve using moving averages of the nearby 10% values), the sums of the shaded areas under the curves represent the lifetime costs and QALE, respectively. As expected, costs dropped after initiation of therapies but increased in final months due to end-of-life care [24]. Lifetime psychometric scores in 2 domains and 4 facets are depicted in **Fig D** in S1 Appendix.

Costs, effectiveness, and ICER of the 192 propensity score-matched patients are summarized in Table 2. Patients receiving afatinib incurred the highest costs in both the progression-

**Table 1. Clinical characteristics of the 192 propensity score-matched patients and those excluded after matching.**

| | Propensity score-matched patients n = 192 | | | | | Excluded patients after matching |
|---|---|---|---|---|---|---|
| | **Afatinib (A)** | **Erlotinib (E)** | **Gefitinib (G)** | **Standardized differences** | | |
| | *n* = 48 | *n* = 48 | *n* = 96 | **A vs. E** | **G vs. E** | *n* = 187 |
| Daily price per person, US$ | 48.2 | 37.0 | 36.3 | | | |
| Age, *n* (%) | | | | | | |
| < 67 years | 32 (66.7) | 31 (64.6) | 60 (62.5) | 0.04 | -0.04 | 120 (64.2) |
| ≥ 67 years | 16 (33.3) | 17 (35.4) | 36 (37.5) | -0.04 | 0.04 | 67 (35.8) |
| Male, *n* (%) | 21 (43.8) | 21 (43.8) | 39 (40.6) | 0 | -0.06 | 63 (33.7) |
| Mutation subtype, *n* (%) | | | | | | |
| Exon 19 deletions | 24 (50.0) | 23 (47.9) | 46 (47.9) | 0.04 | 0 | 70 (37.4) |
| L858R substitution | 22 (45.8) | 24 (50.0) | 49 (51.0) | -0.08 | 0.02 | 91 (48.7) |
| Other mutations | 2 (4.2) | 1 (2.1) | 1 (1.0) | 0.12 | -0.08 | 26 (13.9) |
| Performance status, *n* (%) | | | | | | |
| ECOG: 0–1 | 45 (93.8) | 45 (93.8) | 85 (88.5) | 0 | -0.18 | 168 (89.8) |
| ECOG: 2–4 | 3 (6.3) | 3 (6.3) | 11 (11.5) | 0 | 0.18 | 18 (9.6) |
| Missing | 0 | 0 | 0 | | | 1 (0.5) |
| Disease by recurrence, *n* (%) | | | | | | |
| Recurrent | 9 (18.8) | 8 (16.7) | 12 (12.5) | 0.06 | -0.11 | 45 (24.1) |
| Newly-diagnosed | 39 (81.3) | 40 (83.3) | 84 (87.5) | -0.06 | 0.11 | 142 (75.9) |
| Brain metastasis, *n* (%) | 20 (41.7) | 25 (52.1) | 52 (54.2) | -0.21 | 0.04 | 14 (7.5) |
| PFS, median (IQR) months | 12.3 (7.1–22.2) | 12.7 (6.4–22.0) | 11.5 (8.2–24.3) | | | 12.1 (6.7–23.8) |
| Number of QoLs, *n* | 200 | 194 | 491 | | | 951 |
| Utility value, $\beta_0$ (SE)[a] | 0.80 (0.02) | 0.85 (0.02) | 0.81 (0.02) | | | 0.82 (0.01) |

[a]Intercept of linear mixed model considering random effects from subjects. ECOG: Eastern Cooperative Oncology Group; PFS: progression-free survival

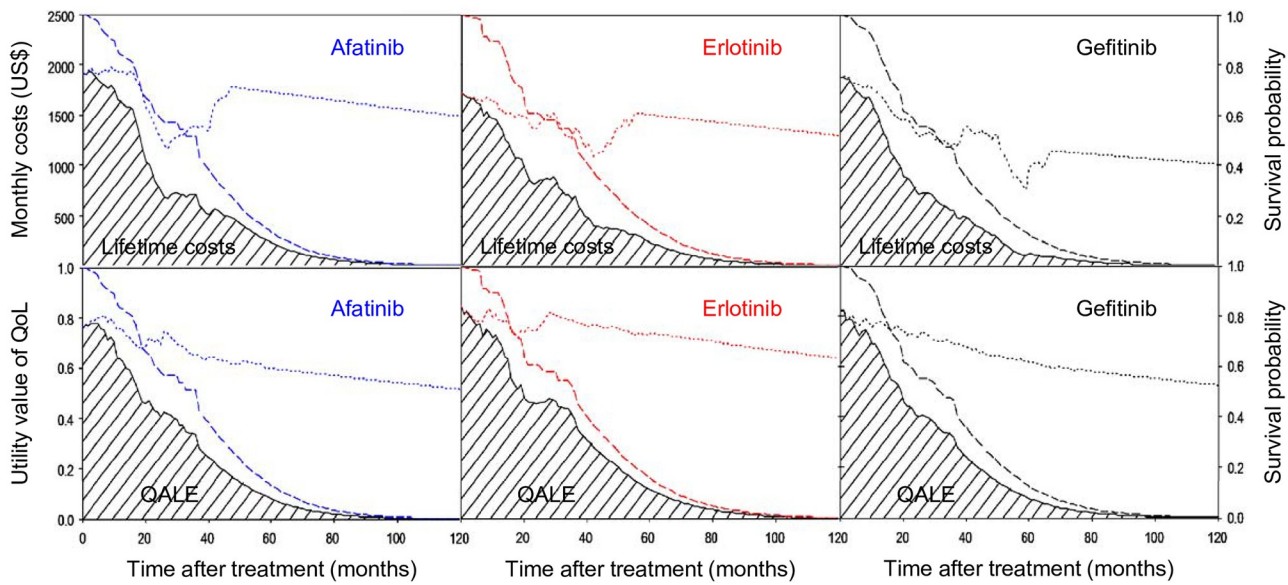

**Fig 1. Lifetime costs and QALE of patients receiving different first-line treatments.** The survival curves (dashed lines), costs and QoL functions (dotted lines); cost- and quality-adjusted survival curves (solid lines) are shown, with the shaded areas representing the lifetime costs and QALE, respectively. 1 US dollar = 29.848 Taiwanese dollars. QALE: quality-adjusted life expectancy; QoL: quality of life.

**Table 2. Costs, effectiveness, and ICER of the 192 propensity score-matched patients.**

|  | Erlotinib | Afatinib | Gefitinib |
|---|---|---|---|
|  | *n* = 48 | *n* = 48 | *n* = 96 |
| Costs, US$ |  |  |  |
| Costs in progression-free period | 31,734 (3,210) | 36,001 (2,874) | 31,873 (2,177) |
| EGFR-TKIs | 19,122 (1,647) | 19,970 (1,519) | 16,546 (1,161) |
| Other than EGFR-TKIs | 12,612 (1,576) | 16,040 (2,052) | 15,328 (1,251) |
| Lifetime costs | 59,005 (3,390) | 64,465 (3,856) | 55,227 (2,249) |
| EGFR-TKIs | 34,693 (2,004) | 34,094 (2,565) | 26,740 (1,246) |
| Other than EGFR-TKIs | 24,360 (1,867) | 30,397 (1,775) | 28,490 (1,564) |
| Chemotherapies | 4,805 (792) | 11,980 (1,212) | 8,683 (844) |
| Lifetime psychometric score, score year |  |  |  |
| Physical | 43.9 (2.3) | 37.1 (1.7) | 37.2 (1.3) |
| Pain | 12.7 (0.6) | 11.1 (0.6) | 11.3 (0.4) |
| Sleep | 10.2 (0.5) | 8.1 (0.3) | 8.8 (0.3) |
| Psychological | 41.7 (2.0) | 35.2 (1.7) | 36.8 (1.2) |
| Bodily appearance | 10.6 (0.6) | 8.9 (0.5) | 9.5 (0.3) |
| Negative feelings | 11.2 (0.6) | 9.7 (0.5) | 10.7 (0.4) |
| Effectiveness |  |  |  |
| Life expectancy, life year | 3.06 (0.14) | 2.94 (0.13) | 2.84 (0.10) |
| QALE, QALY | 2.53 (0.12) | 2.21 (0.11) | 2.20 (0.08) |
| ICER |  |  |  |
| ΔCost /ΔLife expectancy | 17,960 (6,766) | dominated | — |
| ΔCost /ΔQALE | 12,782 (25,001) | dominated | — |

Data presented as mean (standard error) after 100 bootstrap samplings. EGFR: epidermal growth factor receptor; ICER: incremental cost-effectiveness ratio; QALE: quality-adjusted life expectancy; QALY: quality-adjusted life year; TKI: tyrosine kinase inhibitor

free and lifetime periods. Lifetime psychometric scores were lower in the afatinib group, including those in the physical and psychological domains, as well as facet scores of pain, sleep, bodily appearance, and negative feelings. However, the differences in QALE appeared to be modest. The erlotinib group dominated the afatinib group and had an incremental cost-effectiveness of US$17,960/life year and US$12,782/QALY when compared with the gefitinib group.

## Sensitivity analysis

Fig 2 shows the cost-effectiveness scatter plots. A Monte Carlo simulation with 1,000 iterations was conducted to construct the acceptability curves (Fig 3A), which show erlotinib had a probability of 58.6% being cost-effective at a cost-effectiveness threshold of US$24,408 (1 GDP per

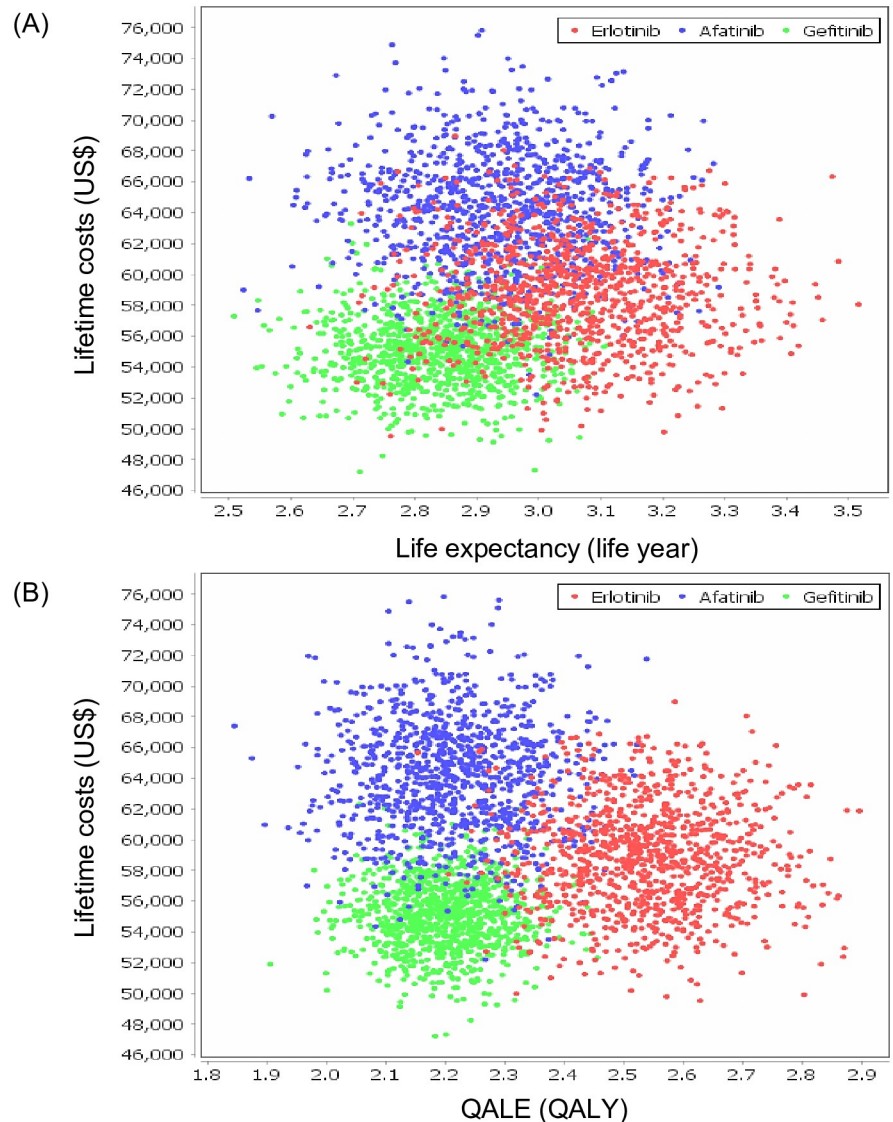

**Fig 2. Cost-effectiveness scatter plots using (A) life expectancy and (B) QALE.** Individual dots represent results after 1,000 iterations. QALE: quality-adjusted life expectancy; QALY: quality-adjusted life year.

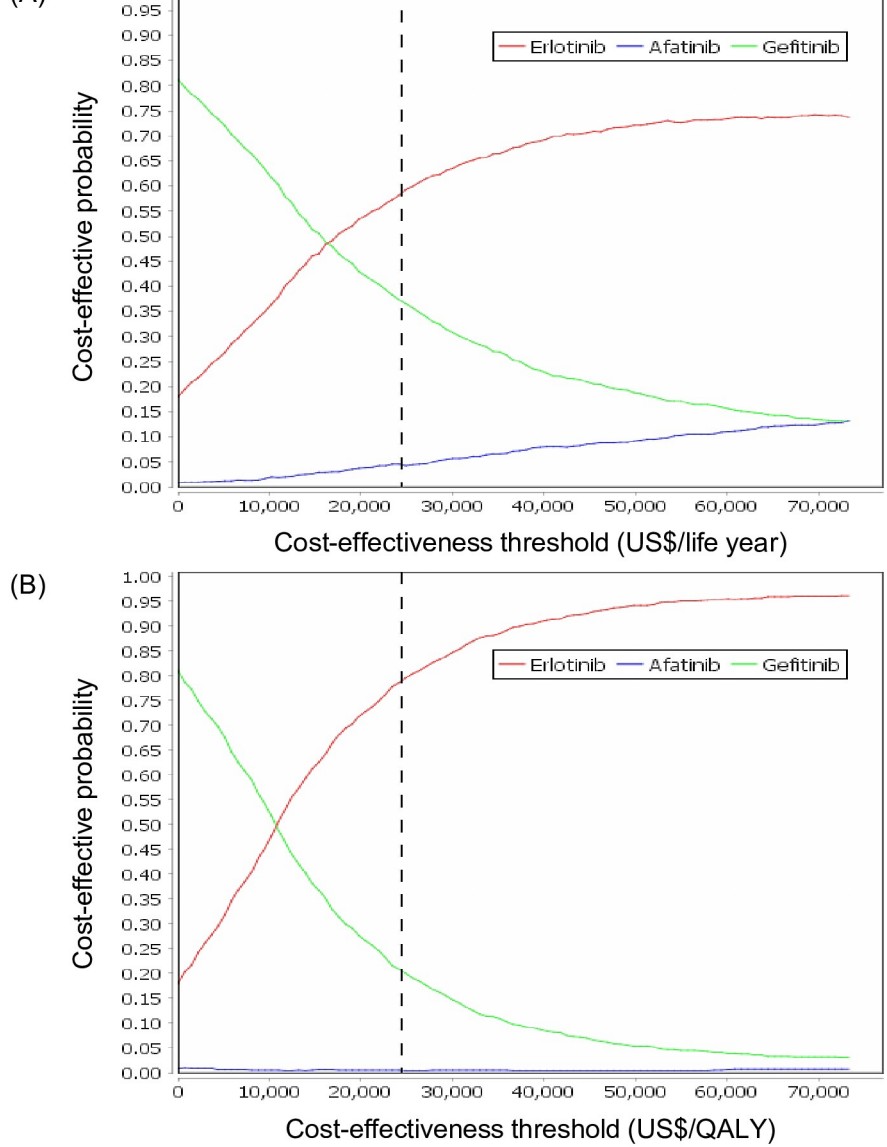

**Fig 3. Acceptability curves of cost-effectiveness thresholds using (A) US$/life year, and (B) US$/QALY.** The dash line represents a threshold of US$24,408 (1 GDP per capita of Taiwan in 2017) / life year or QALY. GDP: gross domestic product; QALY: quality-adjusted life year.

capita of Taiwan in 2017) / life year. If the willingness-to-pay threshold was set at US$24,408 / QALY (Fig 3B), the probability became 78.9%.

We also examined the overall results of the 379 patients before propensity score matching (Table 3).

## Discussion

Most studies analyzed the cost-effectiveness of EGFR testing [25–29] or EGFR-TKI versus chemotherapy as first-line treatment [30–34]. Direct comparisons between different first-line EGFR-TKIs, however, have been performed less frequently. Using OPTIMAL and IPASS trials, Lee *et al.* found that the cost per QALY gained for erlotinib versus gefitinib was US

**Table 3. Costs, effectiveness, and ICER of the 379 patients before propensity score matching.**

| | Afatinib | Erlotinib | Gefitinib |
|---|---|---|---|
| | *n* = 71 | *n* = 57 | *n* = 251 |
| Costs, US$ | | | |
| Costs in progression-free period | 44,785 (3,681) | 30,967 (2,813) | 29,668 (1,177) |
| EGFR-TKIs | 24,992 (2,015) | 18,234 (1,368) | 16,316 (757) |
| Other than EGFR-TKIs | 19,781 (1,850) | 12,736 (1,737) | 13,354 (723) |
| Lifetime costs | 78,612 (6,046) | 62,057 (3,197) | 52,812 (1,343) |
| EGFR-TKIs | 41,823 (3,097) | 35,737 (2,255) | 26,185 (895) |
| Other than EGFR-TKIs | 36,859 (2,852) | 26,338 (2,151) | 26,628 (1,012) |
| Chemotherapies | 11,481 (1,101) | 5,106 (797) | 10,040 (588) |
| Lifetime psychometric score, score year | | | |
| Physical | 44.5 (1.4) | 44.9 (1.9) | 37.2 (0.9) |
| Pain | 13.1 (0.5) | 13.0 (0.6) | 11.1 (0.3) |
| Sleep | 10.1 (0.4) | 10.5 (0.5) | 8.7 (0.2) |
| Psychological | 42.6 (1.5) | 42.8 (1.7) | 36.4 (0.8) |
| Bodily appearance | 10.8 (0.5) | 10.9 (0.5) | 9.4 (0.2) |
| Negative feelings | 11.6 (0.4) | 11.5 (0.5) | 10.2 (0.2) |
| Effectiveness | | | |
| Life expectancy, life year | 3.48 (0.12) | 3.16 (0.14) | 2.79 (0.05) |
| QALE, QALY | 2.61 (0.10) | 2.50 (0.12) | 2.17 (0.05) |
| ICER | | | |
| ΔCost /ΔLife expectancy | 47,765 (26,544) | 24,769 (2,028) | — |
| ΔCost /ΔQALE | 156,385 (55,500) | 31,506 (15,760) | — |

Data presented as mean (standard error) after 100 bootstrap samplings. EGFR: epidermal growth factor receptor; ICER: incremental cost-effectiveness ratio; QALE: quality-adjusted life expectancy; QALY: quality-adjusted life year; TKI: tyrosine kinase inhibitor

$62,419; however, their approach applied an indirect treatment comparison [5]. Similarly, from the experiences of EURTAC and LUX-Lung 3 trials, Ting *et al*. calculated an ICER value of $61,809/QALY for erlotinib versus afatinib via an indirect approach [6]. Recently, Chouaid *et al*. directly compared the cost-effectiveness of afatinib versus gefitinib using LUX-Lung 7 data [7]. Nevertheless, all these studies applied model construction to compare the cost per life year or cost per QALY of 2 EGFR-TKIs [5–7]. In contrast, our study, based on real-world data, directly assessed the lifetime survival, QoL, and medical costs of 3 different treatments along the follow-up courses to estimate the life expectancy, QALE, lifetime psychometric scores, and lifetime costs (Fig 1 and Fig D in S1 Appendix). Our analysis requires fewer assumptions, and the effectiveness and cost-effectiveness estimates produce figures much closer to reality. This study was limited to advanced NSCLC patients with EGFR mutations, and all patients concomitantly using other first-line therapies were excluded. Although instrumental variables can control unobserved covariates [35], such a variable may not easily be found in all study settings. Meta-analysis pooling randomized trials data can avoid confounding [36], but the inclusion and exclusion criteria may limit its application in every day practice. Thus, our method matching each group with propensity scores based on real-world data could still be a viable alternative in providing useful information for allocation of limited resources. We thus tentatively concluded that erlotinib appeared to be cost-effective (Table 2).

Similar to the costs in the progression-free period, more than half of the lifetime costs were attributable to EGFR-TKIs because erlotinib and gefitinib could still be used as the subsequent treatment. That is, the costs of EGFR-TKIs were a major determinant of cost-effectiveness.

During the study period of 2011–2017, generic drugs of gefitinib were not yet available in Taiwan. If the price of afatinib had been reduced 25% to match that of gefitinib (Table 1), it would not have been strongly dominated. Interestingly, the costs of chemotherapies were different among the three groups. However, whether these cost differences are related to the uneven use of subsequent osimertinib remains unknown. Since the out-of-pocket costs of subsequent osimertinib were not recorded in our database, we reviewed the medical records of each patient. Although there was no uneven frequency distribution of the drug use (**Table A** in S1 Appendix [37]: 18.8%, 18.8%, and 16.7% in the afatinib, erlotinib, and gefitinib groups, respectively), the frequencies of positive serum/tissue T790M mutation differed. In contrast to the erlotinib group (18.8%), the afatinib (12.5%) and gefitinib (14.6%) groups had fewer patients with T790M mutation after first-line therapies. Patients without T790M mutation had a shorter progression-free survival using osimertinib [37]; namely, the cumulative use and additional costs of osimertinib would be less. However, after adjusting the additional costs, erlotinib still remained cost-effective.

In accordance with our previous report [12], lower lifetime psychometric scores measured with WHOQOL-BREF for patients receiving afatinib were observed. Lower lifetime scores for pain, bodily appearance, and negative feelings in the afatinib group might result from more severe paronychia, folliculitis, and diarrhea related to afatinib. Nevertheless, more subsequent chemotherapy treatments with increased adverse events in the afatinib group might also contribute to the results. The differences of QALE among three groups were modest, a lower QALE in the afatinib group was not observed.

Several limitations must be acknowledged in this study. First, the work was done in a tertiary hospital in Taiwan, generalizing the results must be cautious. Besides, the spending details on medical services were obtained from the NCKUH database. Because patients might incur expenses outside the hospital, the costs we calculated constitute a conservative estimate. However, we compared the NHI-reimbursed costs in our hospital with the total charges recorded in the NHI database and found that the former accounted for more than 80% of the latter [16], indicating a small bias at most. Second, patients in this study were generally younger and had a better performance status. Thus, the progression-free survival and overall survival would be longer than those excluded, which might lead to an overestimate of effectiveness. Nevertheless, since patients who live longer incur more costs and probabilistic sensitivity analyses accounting for uncertainties showed consistent results, we believe that the estimates would not be overly biased. Third, unobserved prognostic factors including smoking were not considered into propensity score matching because of the lack of data. However, most of EGFR mutation-positive NSCLC patients in Asia (including Taiwan) are never smokers [1], the results would not be biased too much. Fourth, because it is difficult to measure QoL scores close to death, we hypothesized the values beyond the follow-up period to be the same as that near the end of follow-up, which might lead to an over-estimation of QALE. Moreover, numbers of subjects in the afatinib and erlotinib groups were smaller than those receiving gefitinib. Consequently, mean QoL scores and costs after 2 years of follow-up were more easily influenced by outliers. However, as the survival rates after 2 or more years would be low, the bias would not be too big to affect the inference.

This real-world analysis directly compared the effectiveness and cost-effectiveness of three first-line EGFR-TKIs. Erlotinib appeared to be cost-effective from payer's perspective. Lifetime psychometric scores may provide additional information for effectiveness evaluation.

## Supporting information

**S1 Appendix. Supplemental figures and tables to accompany the primary results.**
(DOC)

**S2 Appendix. Anonymized data set for the study findings.**
(XLSX)

**S1 File.**
(PDF)

## Acknowledgments

We are indebted to Yau-Lin Tseng, Yi-Ting Yen, Wen-Ping Su, Shang-Yin Wu, Yu-Ming Yeh, Chien-Chung Lin, and Cheng-Hung Lee for their generous support with the recruitment of subjects. This study is based in part on data from the Cancer Data Bank of National Cheng Kung University Hospital. We are also grateful to Ms. Tzu-I Wu for her statistical analysis and administrative support.

## Author Contributions

**Conceptualization:** Szu-Chun Yang.

**Data curation:** Szu-Chun Yang.

**Formal analysis:** Szu-Chun Yang.

**Funding acquisition:** Szu-Chun Yang.

**Investigation:** Szu-Chun Yang, Wu-Wei Lai, Wu-Chou Su.

**Methodology:** Jung-Der Wang.

**Resources:** Wu-Wei Lai, Jung-Der Wang.

**Supervision:** Jung-Der Wang.

**Visualization:** Szu-Chun Yang.

**Writing – original draft:** Szu-Chun Yang, Jung-Der Wang.

**Writing – review & editing:** Szu-Chun Yang, Wu-Wei Lai, Jason C. Hsu, Jung-Der Wang.

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
