## [Decision Letter · Decision Letter 0]

2 Jan 2020

PONE-D-19-29926

Comparative effectiveness and cost-effectiveness of three first-line EGFR-tyrosine kinase inhibitors: Analysis of real-world data

PLOS ONE

Dear Dr. Yang,

Thank you for submitting your manuscript to PLOS ONE. After careful consideration, we feel that it has merit but does not fully meet PLOS ONE’s publication criteria as it currently stands. Therefore, we invite you to submit a revised version of the manuscript that addresses the points raised during the review process.

In addition to addressing 2 reviewers' comments please address the following comments:

1. Please add Cost-effectiveness Planes (Figures) using your ICERs

2. Please add ICER results in your abstract

3. Please also interprete ICERs in your results section.

We would appreciate receiving your revised manuscript by Feb 16 2020 11:59PM. To enhance the reproducibility of your results, we recommend that if applicable you deposit your laboratory protocols in protocols.io, where a protocol can be assigned its own identifier (DOI) such that it can be cited independently in the future. For instructions see: http://journals.plos.org/plosone/s/submission-guidelines#loc-laboratory-protocols

We look forward to receiving your revised manuscript.

Kind regards,

Khurshid Alam, Ph. D.

Academic Editor

PLOS ONE

Journal Requirements:

3.Please include your tables as part of your main manuscript and remove the individual files. Please note that supplementary tables (should remain/ be uploaded) as separate "supporting information" files

Reviewers' comments:

Reviewer's Responses to Questions

**Comments to the Author**

1. Is the manuscript technically sound, and do the data support the conclusions?

Reviewer #1: Partly

Reviewer #2: Yes

2. Has the statistical analysis been performed appropriately and rigorously? 

Reviewer #1: I Don't Know

Reviewer #2: Yes

3. Have the authors made all data underlying the findings in their manuscript fully available?

Reviewer #1: No

Reviewer #2: No

4. Is the manuscript presented in an intelligible fashion and written in standard English?

Reviewer #1: Yes

Reviewer #2: Yes

5. Review Comments to the Author

Reviewer #1: Cost-effectiveness of drugs using real-world data is truly the area of interest for many researchers/physicians and payers.

However, when using real-world data, the followings must be addressed in the paper to ensure reliability and transparency ,and not to mislead the readers.

Therefore, I suggest to make 3 major revisions and 5 minor revisions for this great work by the team in Taiwan.

Major revisions

・Lack of generalizability must be addressed in Discussion and you need to change the Title to honestly indicate that this work is done in a single tertiary hospital in Taiwan.

・Methods has to be fully revised as there is lack of information. Especially, what information other than QOL and survival are collected prospectively for this research purpose. If you are utilizing the information from EMR which was collected as a part of routine care, it is fine but you need to mention it and how those additional information are abstracted from EMR. It might be helpful to just attach the study protocol approved by IRB before commencement as a supplemental document if you wish to avoid re-writing Method entirely.

・Information is missing for how you handled missing data in QOL or data abstracted from other source which needs to be implemented in either body or supplemental document. Moreover, information about how you summarized QOL data in Figure 1 while every patient have at different timing of visit in real-world setting need to be included.

Minor revisons

・In section of Effectiveness, it is written that the authors determined disease progression based on RECIST v1.1. I wounder if this is true because disease progression is not strictly assessed based on RECIST criteria but also by other evidence such as patient symptoms in real-world setting. Some may decide the switching of therapy due to adverse event too. If the authors wish to mention that the authors reevaluated disease progression by independent review, there needs to be clarification.

・Please make a change to supplemental figure 1 in order to specify how you identify 379 patients from 729 patient overall. you need to clarify the reason of exclusion as it is not clear by just mentioning the detailed information of QOL and cost was provided by 379 patients. (what do you mean by the detailed information?)

・Table 1 has to be explained in detail why these clinical characteristics are chosen for propensity score matching. Smoking status is well known prognostic factor in NSCLC. If information was not collected, you have to mention in Discussion that not all known prognostic factors are adjusted in this research. Moreover, clarification is needed for why these certain comorbidities are selected even though there are several other comorbidities which may affect QOL of NSCLC patient treated with EGFR-TKI such as skin rash.

・It is better to mention each drug costs in Taiwan as Table 2 only show the total healthcare expenditure as you mention afatinib is too expensive in Discussion. Moreover, availability of generic drugs for gefitinib or erlotinib in Taiwan can be included in Discussion.

・Figure 1 is key result for this research, therefore, need to be explained in details. Especially, there seems sudden increase in cost and QOL for afatinib and erlotinib. Also, there may be a over prediction of QoL when reaching to survival probability close to 0.

Reviewer #2: PONE-D-19-29926: Comparative effectiveness and cost-effectiveness of three first-line EGFR-tyrosine kinase inhibitors: Analysis of real-world data

General comments

The research was innovative on methodology in two approaches. First, it used a semi-parametric approach to extrapolate the overall survival (OS) over lifetime. Second, the study patients were matched individually on propensity scores.

The authors claimed the study was based on the real-world dataset of Taiwanese patients that compared treatment effectiveness directly across three TKIs. Regarding the first approach, the OS relied on a statistical model of data projection as the study patients were not followed exhaustively through lifetime. The study justified this approach adequately by citing previous studies for validity of the employed model.

Regarding the second approach, comparability across patients which was based on the propensity-score matching (PSM) across the three TKI groups have not been adequately addressed. To make the three groups comparable, the PSM handles selection bias through the observed covariates. This was justified through balances in the selected 7 covariates shown in Table 1.

The PSM, however, could not control for the unobserved prognostic factors which can lead to omitted-variable bias. An instrumental-variable (IV) technique is a viable alternative to randomization and probably is better than PSM. First, the study should provide adequate argument on how well PSM performed as compared to meta-analysis (direct and indirect treatment comparisons) based on RCTs and pseudo-randomization technique like IV.

The second issue for the second approach is on the study-excluded and included patients. There were 729 patients in total that received the first-line TKIs. Then 350 patients were excluded due to data availability. The remaining 379 patients (52% of 729) were subject to the PSM, where only another 50% (192/379 patients) were successfully matched. The study should shed some lights on the excluded 537 patients in total (74% of 729) that were eligible but did not contribute the analytic dataset.

Other comments about terminology used and reporting/presentation styles are as follows.

Specific comments

Abstract

Introduction:

1. Page 3: “… would improve patients’ values.” The term “patients’ values” is ambiguous; the values on which exact aspects should be pointed out: quality of life, clinical benefits, values for money, or else? This should be more specific.

Main text

Introduction:

2. Page 5: “Randomized trials have shown similar efficacy of these three agents [3, 4].” Reference #3 which was not relevant should not be cited in the present study since it was conducted in patients mixed between 1st line and 2nd line treatments. Reference #4 showed a statistically significant difference in the efficacy measured by PFS, in which afatinib was better than gefitinib (despite non-significant difference in OS because the study was not mature yet).

3. Page 5: “To improve patients’ values, the comparative effectiveness …” The term “patients’ values” again is ambiguous. How the patients’ values could be improved by exploration of the comparative effectiveness and cost-effectiveness of TKIs was unclear?

4.Page 5: “When the cost-effectiveness is considered from the payers’ perspective, real-world data seem

more credible.” Credibility on the baseline survivals or credibility on the comparative effectiveness and cost-effectiveness? The real-world data based on local population tend to be useful for generating the baseline outcomes for the comparator which is least effective such as placebo or chemotherapy. When comparing across innovative drugs (TKIs), the data should be based on the “randomized, controlled” studies, the level-1 evidence. Please provide the references on the evidence that real-world data are more credible from the payer’s perspective.

Methods:

5. Page 6: “This study was approved by the International Review Board of …” This is typo? It should say the Institutional Review Board of …

6. Page 6: “Among 1,828 enrolled cases, 379 patients with EGFR mutation-positive advanced NSCLC under first-line TKIs were abstracted for analysis.”

There were 350 patients not included in the study (Supplementary Table 1, the rightmost column) and 379 patients had detailed information of QoL and costs (shown in Supplementary Figure 1), of which 192 patients were successfully matched. The total of 729 (379+350) patients were relevant to the study and should be mentioned as the enrolled cases instead of all invited lung cancer cases (1,828 patients), which some of them did not receive the EGFR-TKIs.

Otherwise, Supplementary Figure 1 should show the patient flow diagram beginning with a total of 1,828 patients invited, followed by those 1,099 patients who did not receive TKIs were excluded, then 350 patients without information were excluded and 379 remained in the dataset.

The study should elaborate on (1) what were the reasons they were excluded? (2) How did the excluded patients look like and were they much different in characteristics from those included in the study?

To answer (2), the excluded 537 patients in total (in Table 1) should be divided further into two columns separately between 350 patients first excluded (as in Supplementary Table 1) and 187 patients without PSM.

Propensity score matching:

7. Page 6” “To minimize selection bias and ensure better comparability among three different EGFR-TKIs, we used propensity score matching via a greedy algorithm.”

The term “selection bias” should be replaced with more specific terms that the propensity score matching (PSM) aims for and can handle. For example, the PSM is only able accounted for the observed covariates that have been used for determining the probability of obtaining a treatment. However, it is not able to handle selection bias due to the “omitted variables”.

The term “greedy algorithm” should be elaborated technically. For example, how is this method different from conventional approaches in calculating the propensity scores, such as probit or logistic regressions? Does the algorithm perform well for the not so large sample size (379 patients in this study).

QoL and QALE-QALYs

8. Page 7: As the research used both EQ-5D and WHOQOL-BREF for QoL measures, it should be point out which specific measure was used for adjusting the survival years. What are the reasons to analyze the lifetime QoL scores if they were not used for estimating QALE or QALYs.

Otherwise, the study should propose in the beginning the lifetime QoL scores as one of the objectives, say the patients’ values.

Medical costs

9. Page 8: Data on the medical costs were collected until December 2017 for calculating monthly expenditures per case. However, the censor date for survival was in September 2018. Were the medical costs occurring in patients survived during December 2017 to September 2018 not accounted for?

10. Since costs of the TKIs per se played an important role, especially for afatinib in the lifetime costs as shown in the study results. The data on the unit price of each TKI along with the descriptive statistics of utilities per TKI should be shown as the baseline parameters.

Probabilistic sensitivity analysis (PSA)

11. Page 9: “We assumed a gamma distribution for life expectancy, QALE, and costs, …” As healthcare costs are known for very skewed distribution, assuming a gamma distribution of cost data is well accepted. However, a gamma distribution for life expectancy, QALE is not familiar with. References for justifying the distribution of LE and QALE would be helpful.

12. Page 9: “A Monte Carlo micro-simulation with 1,000 iterations was conducted to construct acceptability curves…” A Monte Carlo micro-simulation is too general to be mentioned for the PSA. There were three TKI options to be compared at the same time on cost-effectiveness measures. Specific approaches, either net monetary benefit or net health benefit used for determining the most cost-effective option among the three should be mentioned instead.

Results:

13. The study should present descriptive statistics in the beginning of the Eesults on the unit price of each TKI along with the utilities per TKI as the baseline parameters.

Base-case analysis

14. Table 2: The results on CER should be omitted as the “average ratio” has no implications for policy decision and can create confusions if they were contradicted to the incremental analysis.

Instead, the results on ICER should be calculated hierarchically either by an increasing order in the total costs or by an increasing or in the total effectiveness (LYs, QALE, QALYs).

Based on an increasing order of QALYs, the first ICER should be calculated for afatinib (vs. gefitinib). As erlotinib was more effective and less costly than afatinib, there was no need to calculate the ICER for erlotinib (vs. afatinib) because erlotinib was cost-saving in lay language or technically economic dominance. In this case, it should not say afatinib was least cost-effective. Instead, it should say afatinib was not cost effective since it was just simply dominated by erlotinib and should be dropped totally for the cost-effectiveness comparison.

Then, the ICER for erlotinib (vs. gefitinib) could be calculated and compared with Taiwan’s GDP per capita. In this case, the ICER of erlotinib (vs. gefitinib) was below Taiwanese GDP per capita, hence, the study should conclude that erlotinib was cost effective.

Sensitivity analysis

15. Page 11: Since the PSA compared all three TKIs mutually at the same time, Supplementary Figure 4 should not do a pairwise-plot. Instead, it should plot the scatters for all three TKIs in the same graph by using total cost and total QALYs of each TKI rather than using the incremental values by specifying any TKI as the comparator. In addition, there is no need to draw the CE threshold of Taiwan at this stage.

6. PLOS authors have the option to publish the peer review history of their article (what does this mean?). If published, this will include your full peer review and any attached files.

Reviewer #1: No

Reviewer #2: No

---

## [Author Response · Author response to Decision Letter 0]

30 Jan 2020

Please kindly refer to the Response to Reviewers.

---

## [Decision Letter · Decision Letter 1]

19 Feb 2020

PONE-D-19-29926R1

Comparative effectiveness and cost-effectiveness of three first-line EGFR-tyrosine kinase inhibitors: Analysis of real-world data in a tertiary hospital in Taiwan

PLOS ONE

Dear Dr. Yang,

Thank you for submitting your manuscript to PLOS ONE. After careful consideration, we feel that it has merit but does not fully meet PLOS ONE’s publication criteria as it currently stands. Therefore, we invite you to submit a revised version of the manuscript that addresses the points raised during the review process.

We would appreciate receiving your revised manuscript by Apr 04 2020 11:59PM. To enhance the reproducibility of your results, we recommend that if applicable you deposit your laboratory protocols in protocols.io, where a protocol can be assigned its own identifier (DOI) such that it can be cited independently in the future. For instructions see: http://journals.plos.org/plosone/s/submission-guidelines#loc-laboratory-protocols

We look forward to receiving your revised manuscript.

Kind regards,

Khurshid Alam, Ph. D.

Academic Editor

PLOS ONE

Reviewers' comments:

Reviewer's Responses to Questions

**Comments to the Author**

1. If the authors have adequately addressed your comments raised in a previous round of review and you feel that this manuscript is now acceptable for publication, you may indicate that here to bypass the “Comments to the Author” section, enter your conflict of interest statement in the “Confidential to Editor” section, and submit your "Accept" recommendation.

Reviewer #1: (No Response)

2. Is the manuscript technically sound, and do the data support the conclusions?

Reviewer #1: Partly

3. Has the statistical analysis been performed appropriately and rigorously? 

Reviewer #1: Yes

4. Have the authors made all data underlying the findings in their manuscript fully available?

Reviewer #1: Yes

5. Is the manuscript presented in an intelligible fashion and written in standard English?

Reviewer #1: Yes

6. Review Comments to the Author

Reviewer #1: As for the my previous comment 5, you included the detailed explanation regarding the reason why 350 patients were excluded from this research in the manuscript and supplemental figure 1.

If the reason for exclusion of 346 out of 350 patients is no informed consent obtain for this research, then it is not appropriate to use their data and summarize in Table 1 from the ethical point of view.

Please delete all relevant data from patients without signed informed consent in this manuscript.

7. PLOS authors have the option to publish the peer review history of their article (what does this mean?). If published, this will include your full peer review and any attached files.

Reviewer #1: No

---

## [Author Response · Author response to Decision Letter 1]

20 Feb 2020

Please kindly refer to the Response to Reviewers. Thank you!

---

## [Decision Letter · Decision Letter 2]

24 Mar 2020

Comparative effectiveness and cost-effectiveness of three first-line EGFR-tyrosine kinase inhibitors: Analysis of real-world data in a tertiary hospital in Taiwan

PONE-D-19-29926R2

Dear Dr. Yang,

We are pleased to inform you that your manuscript has been judged scientifically suitable for publication and will be formally accepted for publication once it complies with all outstanding technical requirements.

With kind regards,

Khurshid Alam, Ph. D.

Academic Editor

PLOS ONE

Additional Editor Comments (optional):

Reviewers' comments:

Reviewer's Responses to Questions

**Comments to the Author**

1. If the authors have adequately addressed your comments raised in a previous round of review and you feel that this manuscript is now acceptable for publication, you may indicate that here to bypass the “Comments to the Author” section, enter your conflict of interest statement in the “Confidential to Editor” section, and submit your "Accept" recommendation.

Reviewer #1: All comments have been addressed

2. Is the manuscript technically sound, and do the data support the conclusions?

Reviewer #1: Partly

3. Has the statistical analysis been performed appropriately and rigorously? 

Reviewer #1: Yes

4. Have the authors made all data underlying the findings in their manuscript fully available?

Reviewer #1: Yes

5. Is the manuscript presented in an intelligible fashion and written in standard English?

Reviewer #1: Yes

6. Review Comments to the Author

Reviewer #1: I believe all the comments have been addressed. I have no objection to accept this manuscript for PLOS ONE journal.

7. PLOS authors have the option to publish the peer review history of their article (what does this mean?). If published, this will include your full peer review and any attached files.

Reviewer #1: No

---

## [Editor Report · Acceptance letter]

25 Mar 2020

PONE-D-19-29926R2 

Comparative effectiveness and cost-effectiveness of three first-line EGFR-tyrosine kinase inhibitors: Analysis of real-world data in a tertiary hospital in Taiwan 

Dear Dr. Yang:

I am pleased to inform you that your manuscript has been deemed suitable for publication in PLOS ONE. Congratulations! Your manuscript is now with our production department. 

With kind regards,

on behalf of

Dr. Khurshid Alam 

Academic Editor

PLOS ONE